# Ophthalmic Complications After Dental Procedures: Scoping Review

**DOI:** 10.3390/diseases13080244

**Published:** 2025-08-04

**Authors:** Xingao C. Wang, Cindy Zhao, Kevin Y. Wu, Michael Marchand

**Affiliations:** 1Faculty of Medicine and Health Sciences, McGill University, Montreal, QC H3G 2M1, Canada; 2Faculty of Dentistry, University of Montreal, Montreal, QC H3T 1J4, Canada; 3Department of Surgery, Division of Ophthalmology, University of Sherbrooke, Sherbrooke, QC J1K 2R1, Canada; yang.wu@usherbrooke.ca; 4Department of Dentistry, Faculty of Dental Medicine and Oral Health Sciences, McGill University, Montreal, QC H3A 0G4, Canada

**Keywords:** dentistry, ophthalmology, complications, anesthesia-induced complications, multidisciplinary management

## Abstract

**Introduction:** Ocular complications associated with dental procedures are diverse but have been primarily reported through case reports and series, with no comprehensive reviews to date. The underlying mechanisms of these complications are often poorly understood by medical professionals, partly due to limited interdisciplinary education. This review aims to bridge this gap by summarizing the relevant anatomical connections between the oral and ocular regions, exploring the mechanisms through which dental procedures may lead to ophthalmic complications, and detailing their clinical presentations, progression, and potential management and preventive strategies. **Methods:** Published case reports and case series from 1950 to October 2024 that described ophthalmic complications in human patients following dental procedures were included in this scoping review. **Results:** Dental procedures can give rise to a variety of ophthalmological complications, whether neuro–ophthalmic (e.g., diplopia, ptosis, or vision loss), vascular (e.g., retrobulbar hemorrhage or cervical artery dissection), infectious (e.g., orbital cellulitis or abscess), mechanical (e.g., orbital trauma or fractures), or air-related (e.g., orbital and subcutaneous emphysema). **Conclusions:** Most of the ophthalmological complications following dental procedures are often reversible, but some can be vision-threatening or lead to permanent sequelae if not promptly recognized and managed. Prevention through precise technique and anatomical awareness, early identification of symptoms, and timely multidisciplinary collaboration are crucial to minimizing risks and ensuring better patient outcomes.

## 1. Introduction

Dentistry has undergone significant evolution over time, marked by the introduction of new techniques and advancements [1]. However, these advancements have not eliminated the risk of complications, some of which can carry significant morbidity. Among these, ocular damage—potentially leading to permanent vision loss—is a rare but serious complication [2]. Reports of ophthalmological complications stemming from dental procedures have been documented since the mid-1900s, with their frequency increasing in recent decades [2].

This review aims to identify the various ophthalmic complications that can arise from dental procedures and the mechanisms that lead to such complications. While a literature review by Cody Lo et al. in 2021 focused specifically on acute visual loss [2], our review takes a broader approach, encompassing a wider spectrum of ocular and adnexal complications. This information is relevant to a diverse audience, including ophthalmologists, dentists, and non-ophthalmologists such as emergency physicians. Previous surveys highlight a gap in formal cross-disciplinary education, with many healthcare professionals lacking training in fields outside their own specialty [3]. For instance, dentists may have limited medical education in ophthalmology, while physicians often lack in-depth knowledge of dental procedures [3]. By fostering a better understanding of the relevant anatomy and the underlying mechanisms behind these complications, this review aims to bridge this knowledge gap and ultimately improve patient care.

## 2. Materials and Methods

We performed a search of bibliographic databases, including PubMed and Embase, to identify relevant articles discussing ophthalmic complications associated with dental procedures. No formal protocol was developed for this review. Our search spanned from January 1950 to December 2024. Search terms included variations of “ophthalmology,” “ocular complications,” and “vision,” combined with terms such as “dentistry,” “dental procedures,” and “dental anesthesia” to capture the relevant literature. The Boolean operator AND was used to combine terms from each category to identify studies discussing both ophthalmology and dentistry. An example search query was: “ocular complications” AND “dental procedures.” The most recent search was performed on 10 July 2025. This review was performed under PRISMA guidelines. 

Articles were screened based on their titles and abstracts for relevance. The inclusion criteria were as follows:
Articles published in English.Studies involving human subjects.Articles discussing any ophthalmic complications following dental procedures.

The term ophthalmic complications referred to a large range of ocular complaints, including, but not limited to, amaurosis, vision loss, diplopia, ophthalmoplegia, nerve palsy, artery complications, endophthalmitis or other infectious complications, globe penetration, cycloplegia, mydriasis, and ptosis.

To enhance methodological rigor, two independent reviewers (K.W. and X.W.) conducted the search and screened articles for inclusion. The reference lists of articles obtained were consulted to retrieve cross-references. Studies citing these articles were also reviewed. A flow diagram (Figure 1) illustrates the article search and selection process.

From the final list of eligible articles, the following data were manually extracted (for further details, refer to the Excel file in the Appendix A):
Study type.Dental procedures associated with complications.Time of onset from procedure to complication.Proposed mechanisms.Treatments and outcomes.Number of cases.Other significant details.

The majority of our sources are case reports, and as such, formal critical appraisal was not applicable. Data were grouped by etiology of ophthalmic complications, including neuro–ophthalmic, orbital and periorbital, vascular, traumatic, and other complications. 

## 3. Results and Discussion

### 3.1. Epidemiology and Case Statistics

Ophthalmic complications following dental procedures are rare. For example, the prevalence of ophthalmic complications is less than 0.1% following dental anesthesia [4]. Of these, most are documented in the form of case reports and small case series. In this review, a total of 85 cases of ocular complications after dental procedures were identified, with the most common complication being neuro–ophthalmic in nature. Within neuro–ophthalmic complications, 20 out of 85 patients reported diplopia, followed by 19 out of 85 experiencing vision loss, and a rare few had ptosis or loss of accommodation (6 out of 85). Orbital and periorbital complications followed, with 11 out of 85 cases of orbital cellulitis and abscess, which mostly occurred after maxillary molar extraction. Eight out of eighty-five also reported orbital emphysema, most commonly due to high-pressure dental devices, and seven out of eighty-five developed periorbital necrotizing fasciitis. Finally, vascular complications, periorbital blanching, and orbital trauma/fractures were amongst the rarest forms of complications post-dental procedure, along with one case of facial nerve palsy. 

### 3.2. Anatomical and Physiological Connections

The anatomical and physiological connections between the orbit and oral cavities are significant due to their shared vasculature, innervation, and physical proximity.

#### 3.2.1. Vascular Connections

The vascular supply to the orbital and oral cavities arises, respectively, from the internal and external carotid arteries, which branch from the common carotid artery bifurcation [5]. The external carotid artery gives rise to the internal maxillary artery, a vessel with considerable variation in topography whose position and diameter are relative to the mandibular foramen [6]. This anatomical variability increases the risk of accidental arterial penetration during dental procedures, such as local anesthesia administration [6]. Anesthetic agents inadvertently introduced into the inferior or posterior superior alveolar arteries may travel in a retrograde fashion through the internal maxillary, middle meningeal, and ophthalmic arteries if sufficient injection pressure is applied (Figure 2) [7]. This retrograde flow can lead to local vasoconstriction and ischemia of orbital structures. Through anastomoses, the ophthalmic and middle meningeal arteries supply the central retinal and lacrimal arteries, which underscores the potential for dental anesthesia to impact ocular function if improperly administered [7,8].

Moreover, the middle meningeal artery can supply cranial nerves IV and VI [9]. Notably, in 4% of patients, the ophthalmic artery originates from the middle meningeal artery rather than the internal carotid artery [10]. Similarly, in 3.5% of individuals, the ophthalmic branch of the middle meningeal artery provides the primary blood supply to the lacrimal artery [9]. These anatomical variations create direct pathways for local anesthetics to reach ophthalmic arteries in cases of arterial injection. Variations in the inferior alveolar, internal maxillary, mental, and lingual nerves may also contribute to inadvertent ophthalmic complications [11]. 

The viability of tissues supplied by the ophthalmic artery varies depending on the extent of collateral circulation. For instance, the presence of a cilioretinal artery in 15–20% of the population can positively influence the outcome of retinal artery occlusion by providing an additional independent arterial supply to the retina [9].

#### 3.2.2. Venous Connections

The oral cavity primarily drains into the pterygoid venous plexus, which connects to the cavernous sinus via small emissary veins traversing the foramina rotundum, lacerum, and ovale. The cavernous sinus contains cranial nerves III, IV, V1, V2, and VI, as well as the internal carotid artery and sympathetic fibers (Figure 3) [6]. Substances or pathogens entering the cavernous sinus can lead to various ocular complications [6,7]. The absence of valves in the veins of the head and neck allows for local anesthetics to flow along multiple pathways depending on pressure gradients. Extensive venous anastomoses provide numerous routes for anesthesia to reach the orbit [12]. 

#### 3.2.3. Neurological and Pathological Pathways

Nerve involvement near the inferior orbital fissure can affect the extraocular muscles, orbital tissues, and nerves due to their proximity [6,13]. Pathogens may travel from an extracted tooth through the posterior maxilla, infratemporal fossa, inferior orbital fissure, and, ultimately, the subperiosteal orbital region. This migration can occur via venous anastomoses, neurovascular foramina, and congenital or acquired bony dehiscences [14]. The lack of a bony barrier between the orbital and retromaxillary regions facilitates access to the inferior orbital fissure through the pterygopalatine fossa. This mechanism is implicated in complications following greater palatine canal or posterior superior nerve blocks [4,13]. The pterygoid venous plexus also connects directly to the orbit via the inferior orbital fissure [12]. 

#### 3.2.4. Additional Pathways

Ocular complications can arise through several additional mechanisms. For instance, microorganisms may invade the orbit via the sinuses following posterior maxillary tooth extraction [15]. Local anesthetics deposited near the pterygoid canal or within the pterygomaxillary fossa can diffuse to the orbit [16,17]. High-speed air-driven instruments may introduce air into the subcutaneous tissue planes of the face, potentially reaching the periorbital tissue through the pterygomaxillary region during upper third molar extraction [18]. Furthermore, the roots of mandibular molars communicate directly with the sublingual and submandibular spaces, which are continuous with the pterygomandibular, parapharyngeal, and retropharyngeal spaces. Air introduced during dental procedures may travel along these planes [18,19]. Local anesthesia administered to the upper or lower maxillary regions can diffuse into the stellate ganglion via the pterygomandibular space and fascial planes, potentially causing Horner’s syndrome [17,18]. 

### 3.3. Neuro–Ophthalmic Complications

#### 3.3.1. Diplopia

Diplopia resulting from dental procedures is exceedingly rare but is often immediately recognized by patients [4]. Maxillary injections are more commonly associated with diplopia than mandibular injections [20]. The literature suggests that differences in bone density between the two maxillae may explain the difference in rates of ophthalmic complications [21]. 

The lateral rectus muscle is particularly vulnerable because of the long intracranial course and anatomical position of the abducens nerve [22]. Of the reported cases, eleven out of twenty involved inadequate eye abduction, lateral rectus muscle palsy, or sixth cranial nerve paresis; one case involved oculomotor palsy; one case involved trochlear palsy; one case involved both trochlear and oculomotor palsy; one case involved total ophthalmoplegia; and five cases did not specify the type of diplopia [4,11,17,20,21,22,23,24,25,26,27,28,29,30,31,32,33]. 

Diplopia is often linked to the anesthetic administration method and its ability to diffuse into adjacent structures. In the mandible, diplopia typically occurs after an inferior alveolar nerve block or a Gow-Gates mandibular injection [6,7,11,12,23,25,27,30,34]. In the maxilla, it is associated with posterior superior alveolar nerve blocks, greater palatine nerve blocks, or maxillary nerve blocks administered via the greater palatine canal (see Figure 4 for sites of injection) [6,13,17,20,22,29,31,33]. Lidocaine and articaine are the two most used anesthetics in reported cases. The higher incidence of ophthalmic complications with articaine may be attributed to its superior diffusion properties [12,25]. 

The onset of complications generally occurs immediately or within minutes after the procedure [4,11,12,17,20,21,22,23,25,26,27,29,30]. The underlying vascular mechanisms include intravenous or intra-arterial injection, which can lead to retrograde anesthetic flow to the cavernous sinus, where the third, fourth, and sixth cranial nerves are located, or to the orbit via the middle meningeal artery (Figure 3) [4,6,9,18,20,22,23,25,28,31,35]. The anesthetic may also diffuse directly through bony pathways, such as the inferior orbital fissure or the pterygopalatine fossa, affecting specific ocular muscles like the lateral rectus, or causing ischemia of the lacrimal artery that supplies the muscle [4,22]. 

Periorbital blanching has been reported after dental anesthesia in the maxilla, with some isolated cases where it occurs concurrently with diplopia [26,30,31,33]. This phenomenon likely results from arterial wall damage leading to vasospasm within the internal carotid plexus or the ophthalmic artery [26]. In rare instances, diplopia has been caused by intraorbital hematoma. Baba et al. (2017) reported a case in which diplopia developed due to a post-operative hemorrhage following the extraction of a maxillary third molar [36]. The bleeding extended into the intraorbital region from the pterygomaxillary and infratemporal spaces via the inferior orbital fissure. Recovery in this case exceeded one week, although most cases resolve spontaneously within minutes to hours [4,6,11,12,17,20,22,23,25,26,27,28,34]. Management recommendations include covering the affected eye to address binocular diplopia and providing support to patients while adjusting to temporary monocular vision [4,12,17,22,27]. Preventive measures involve aspirating in two planes and injecting slowly, while carefully considering anatomical structures relevant to the injection technique [17]. It is essential to recognize that unresolved and painful diplopia may indicate cavernous sinus syndrome or Tolosa–Hunt syndrome, necessitating further diagnostic investigation [24]. 

#### 3.3.2. Ptosis and Loss of Accommodation

Ptosis and loss of accommodation are presentations that can often present alongside diplopia, and the physiology behind such pathology remains similar. There are three documented cases of ptosis, with two classified as Horner’s syndrome [37,38]. These young and healthy female patients presented with classical ptosis and miosis, with one patient having additional diplopia, suggesting that the anesthetic agent could have possibly infiltrated the cavernous sinus, affecting the sympathetic plexus [38]. Other mechanisms include arterial infiltration or direct diffusion through the pterygomaxillary fossa [37,39,40]. Another hypothesis considers possible anatomic variations of the branching of the ophthalmic artery from the middle meningeal artery instead of the internal carotid artery [41]. In the context of a posterior superior alveolar nerve block, the infiltration of anesthetic from the posterior superior alveolar artery to the middle meningeal artery, connected to the ophthalmic and lacrimal arteries, is possible. Finally, three out of six patients had loss of accommodation secondary to ciliary spasm [41] and ciliary palsy [42]. Two of these cases occurred during inferior alveolar blocks [42]. Similar to previous hypotheses, anesthetic toxicity (e.g., lidoaince) or its injection into the neurovascular bundle or retrograde flow to the cavernous sinus [42,43]. Ptosis and loss of accommodation overlap between many clinical diagnoses, such as Horner’s syndrome and ciliary palsy. A comprehensive understanding of the anatomy connecting the oral cavity to the orbit is essential in reducing the risk of these complications. Moreover, two-plane aspiration should always be performed to minimize intra-arterial anesthetic infiltration. Despite this, if a patient develops signs of ptosis or loss of accommodation likely secondary to anesthetic infiltration, its effect is likely temporary, lasting the expected time of the anesthetic agent itself. 

#### 3.3.3. Vision Loss

Vision loss is a notable ophthalmic complication associated with dental procedures, encountered more frequently during mandibular dental procedures compared to diplopia, which is predominantly observed following maxillary ones [4,7,20,44]. Additionally, men appear to be more susceptible to this complication, whereas women are more likely to experience diplopia. Notably, 13 out of 19 reported cases involved male patients, most of whom had no relevant medical history [7,8,10,16,23,44,45,46,47,48,49,50,51,52]. Interestingly, two of the nineteen cases reported diplopia as a concurrent ocular complication with vision loss [51]. The anesthetic most likely involved the abducens, oculomotor, and optic nerves, which would explain the concomitant ocular complications [51]. 

The clinical diagnosis may include diminished visual acuity, flame-shaped hemorrhages, a pale optic disk, and, in specific cases like central retinal artery occlusion, a cherry red spot [10,23,46]. Most cases of vision loss have been associated with local dental anesthetics, particularly inferior alveolar nerve blocks [8,44,50,51,52]. An intriguing case reported by Parc et al. (2004) suggested that tooth extraction and subsequent debris might have caused optic nerve ischemia, even in the absence of local anesthesia [45]. 

Similarly, Kravitz et al. (2019) described vision loss following intravenous sedation and local anesthesia during a dental extraction, hypothesizing potential etiologies such as local anesthesia, IV sedation, or embolism of dental debris [16]. In contrast, Harvey (2019) proposed that intravenous sedation-induced hypotension led to compensatory vasodilation, which increased the risk of nonarteritic anterior ischemic optic neuropathy (NAION) through mechanisms like venous occlusion or arteriolar compartment syndrome [53]. Optic nerve ischemia is due to a decreased perfusion to the optic nerve and leads to vision loss. Cases of anterior ischemic optic neuropathy (AION) have been identified, with the majority linked to cilioretinal artery occlusion, posterior ciliary arteries, or vasoconstriction and vasospasm [10,16,48,49,54]. One case of branch retinal artery occlusion (BRAO) has been documented and resulted in permanent vision loss [46]. Additionally, a case of central retinal venous occlusion (CRVO) has been reported [47]. Notably, the vascular pathways involved in optic nerve ischemia and retinal artery occlusions from dental procedures share similar mechanisms and pathways to those observed in diplopia and ptosis [7,44,45,47]. 

Vision loss may also occur secondary to other complications arising from dental procedures, such as internal carotid artery dissections or orbital cellulitis, which are discussed further in this article. Typically, the onset of vision loss is immediate or occurs within 1–2 days following the procedure. Outcomes vary widely, ranging from transient episodes lasting up to 10 months to permanent vision loss, depending on the severity of the ischemic event. Of the 19 cases, 15 were transient. Severe ischemic events often result in permanent vision impairment [7,8,10,16,23,44,45,46,47,49,50,52]. Early intervention improves recovery prospects by mitigating orbital tissue damage and restoring perfusion [35,46]. For instance, one case documented an attempted ocular massage 5 h after the complication, which proved to be ineffective, possibly due to treatment delay [10]. 

Prevention strategies for vascular complications align closely with those recommended for diplopia and ptosis.

#### 3.3.4. Cervical Artery Dissection

Cervical artery dissection during dental procedures, although rare, can lead to severe neurological, ophthalmologic, and cardiovascular complications. It is an etiology of Horner’s syndrome [5,55,56,57,58]. Dental procedures involved in the dissection of the cervical artery include dental abscess removals, tooth extractions, and nerve blocks [55,56,57]. Other contributing risk factors involve prolonged neck hyperextension during dental procedures and periodontal infections. Both elements can, respectively, promote tearing or weakening of the arterial walls [55,57,58]. In fact, the prolonged hyperextension of the neck may subject cervical arteries to forces of stress that lead to their trauma [58]. Furthermore, Delgado et al. (2015) described a case where the microbes in a patient’s periodontal infection and her body’s immunological response to it could have induced the degradation of her carotid arterial wall [55]. The onset of complications varies from immediate manifestation to a week after the dental treatment [55,56,57,58]. Molad et al. (2016) state that cervical artery dissection may present non-specific symptoms or be asymptomatic [58]. However, according to a case series of 146 patients with internal carotid artery dissection, the combination of painful Horner’s syndrome with orbital pain and ipsilateral head or neck pain with quick onset is extremely characteristic of internal carotid artery dissection [5]. Moreover, it stated that oculomotor nerve palsies, although rarely occurring, are always associated with other symptoms of dissection, like pain, Horner’s syndrome, or visual loss. The mechanism of the palsies can be attributed to the dissected artery directly compressing the ocular motor or the deterioration of blood supply quantity to the cranial nerves from the internal carotid artery [5]. It is important to highlight that cervical artery dissection is one of the most common causes of ischemic stroke in young adults [5,57,58]. Prompt diagnosis and appropriate management are essential to prevent further cerebral and ocular complications [5]. Strict bed rest coupled with anticoagulation therapy, which includes heparin in the acute phase and warfarin in the following months, will prevent subsequent carotid thrombosis and embolism [5,55,56,57,58]. Most patients who received treatment after dissections during dental work experienced slight to no residual ocular problems within a few months [55,56,58]. 

### 3.4. Orbital and Periorbital Complications

#### 3.4.1. Orbital Cellulitis and Abscess

Orbital cellulitis is most often associated with sinus infections and presents with fever, swelling, and erythema. Post-septal cellulitis/abscess will lead to additional clinical symptoms such as proptosis, chemosis, limited EOM, and possible change in visual acuity. In the context of dental procedures, maxillary molar extraction was the most common cause of orbital cellulitis and abscess [14,15,59], with one case following an oroantral communication repair [60]. The development of complications occurred several hours to days after the procedure. Of these cases, a minority of patients were found to have an underlying active infection at the time of the procedure, which could have increased the risk of infection. These include pulpitis and periapical periodontitis [61], as well as extraoral infections such as maxillary sinus infection [59] and URTI [61]. Close to all patients had proptosis and chemosis, with four out of eleven patients noting visual acuity changes [15,59,61]. 

The close anatomical relationship between maxillary molar roots and the maxillary sinus allows for an infection to spread to the orbit through bony erosion of the orbital floor, the ethmoid sinus, or infraorbital canal [15,62]. From the maxillary sinus, there is also a direct extension to the inferior orbital fissure through the infratemporal and pterygopalatine fossa [60]. Soft tissue and hematogenous spread are also possible [14,61,63]. 

Most often, these cases were treated with intravenous antibiotics, steroids, and surgical exploration and drainage [15,59,60]. CT was the most common imaging modality. Emergency decompression was also performed in cases where damage to the optic nerve was suspected [15,59]. Furthermore, prompt diagnosis and treatment are essential to prevent further complications such as cavernous sinus thrombosis, which requires additional anticoagulant therapy [64,65], and endogenous endophthalmitis, which may lead to pars plana vitrectomy [66]. 

Currently, routine prophylactic antibiotics following dental procedures are not recommended considering low-certainty evidence in healthy patients and the increase in antibiotic-resistant organisms [67]. However, several of the discussed cases involved a pre-existing oral or extra-oral infection, and it is unclear if these infections were direct risk factors for the orbital abscess and cellulitis. Future epidemiological research on this subject will allow for clarifications and ultimately guide recommendations to reduce the risk of secondary ocular complications.

#### 3.4.2. Orbital and Subcutaneous Emphysema

##### Orbital Emphysema

Orbital emphysema is often a benign complication following trauma to the orbital bones. However, a few instances of orbital emphysema secondary to dental procedures have been documented, with the most common being tooth extractions [68,69,70], followed by endodontic treatment [71], root canal treatment [18], and dental implants [72]. Orbital emphysema is usually unilateral with crepitus upon palpation of the affected area; may extend beyond the orbit to the neck, mandible, and cheek; and can appear immediately or days following the procedure. In a previously healthy individual, the mechanism commonly involves inevitable damage to the mucosa during dental procedures combined with the use of high-pressure dental instruments [73]. Importantly, the involvement of the orbit can lead to severe and life-altering consequences such as optic nerve ischemia, compression, and risks of embolisms considering its proximity to the cavernous sinus [74].

##### Subcutaneous Emphysema

Subcutaneous emphysema occurs mostly following molar extraction, as the roots of the molars are directly linked to the sublingual and submandibular space; thereafter, it can travel into the infraorbital and orbital spaces through fascial spaces of the head and neck [70] or bony defects [72]. Normal saline used during dental procedures has also been documented to cause edema around the eye through a similar mechanism [18]. Damage to the mucosa is mostly linked to the dental procedure itself, but there is a documented case of subcutaneous facial emphysema in a 6-year-old with an odontogenic infection who was set to undergo tooth extraction. The emphysema occurred immediately following injection of local anesthesia, during which a high-pressure air device was inside the oral cavity to distract the child from the injection [75]. This case revealed the possible risk of emphysema through the anesthetic site or through the already damaged mucosa due to infection. Nonetheless, this allows for air to be introduced through the damaged mucosa into facial planes. 

Most cases of orbital and subcutaneous emphysema are resolved with time, close follow-ups, and antibiotics. Important complications to keep in mind include intrathoracic emphysema, pneumomediastinum, and pneumothorax [71,75]. Finally, odontogenic infection caused by gas-forming anaerobes remains an important and possibly life-threatening differential to keep in mind when a patient presents with subcutaneous emphysema, as it has been documented to lead to fatal mediastinitis [76]. In order to prevent emphysema in the context of dental procedures, the use of rubber dams and remote exhaust or electrical motor-driven instruments is preferred [71]. Valsalva maneuvers should also be avoided in the early stages of recovery. If clinical presentation is suggestive of orbital involvement or intraocular pressure, timely decompression is required. Finally, CT scans of the affected region can be performed to evaluate the extent of emphysema and differentiate it from other causes, such as infection. CT scans of the chest should be performed when pneumomediastinum is suspected (e.g., dyspnea, back pain, Hamman’s sign) [69].

#### 3.4.3. Periocular Necrotizing Fasciitis

Periocular necrotizing fasciitis is a rare condition given that necrotizing fasciitis rarely occurs in the facial region. The most common cause is dental infection [77]. However, there are a few reported cases where dental procedures, i.e., root canal treatments, the extraction of posterior teeth, and abscess draining, have led to periocular necrotizing fasciitis [77,78,79,80,81,82,83]. Clinical presentation initially comprises pain and swelling or cellulitis [77,78,79,80,81,82,83]. Although there exist clinical features that differentiate necrotizing fasciitis from other conditions, the lack of pathognomonic signs can lead to a misdiagnosis, which can worsen prognosis [82]. Bhaskaran et al. (2019) documented a case where a patient was misdiagnosed twice following a root canal procedure of her upper right premolar. Initially, the pain and swelling of her cheek were attributed to post-endodontic side effects, then to angioedema. It was only upon hospital admission that surgical debridement was performed, revealing necrotic tissue [79]. Of the reported cases, five out of seven involved women, and one case was unspecified. All patients were above the age of 18. Most did not have relevant medical history. The onset of complications typically occurs within 24 h to 2 weeks after the procedure, with most cases being unilateral [77,78,79,80,81,82,83]. 

The exact pathways leading to periocular necrotizing fasciitis are not clearly defined in the literature. However, it is suspected that the dental treatment performed either led to a new infection or exacerbated a pre-existing one. It then spread through deep fascial planes to the orbit and periocular structures [83]. This suggests that the necrosis often spares the overlying skin in the early stages of this condition, which can hinder prompt diagnosis. 

Prognosis is closely related to the promptness of the treatment and the extent of the infection. Hence, early diagnosis and therapeutic management are critical. These include aggressive surgical debridement, removal of necrotic tissue, and immediate broad-spectrum antibiotic administration while awaiting bacterial culture results [77,78,79,80,81,82,83]. Compared to the timing and severity of the necrotizing fasciitis following a dental procedure, the influence of predisposing health conditions on prognosis remains unclear. For instance, Shindo et al. (1997) reported a case involving a patient with diabetes mellitus who recovered despite having residual enophthalmos and ptosis [83]. In contrast, Clement and Hassall (2004) documented a patient with end-stage renal failure who required globe removal due to extensive necrosis [82].

### 3.5. Vascular Complications

#### Retrobulbar and Subretinal Hemorrhage

Retrobulbar hemorrhage is known as an ocular emergency due to its potential rapid progression to orbital compartment syndrome. Clinical presentation includes proptosis, chemosis, and elevated intraocular pressure. When visual acuity or pupils are affected, emergency lateral canthotomy and cantholysis are performed. Bleeding in the retrobulbar space is often associated with trauma. However, three cases of retrobulbar hemorrhage following dental extraction are documented [84,85,86]. One of the three patients had hypertension and diabetes mellitus [86], while the others were young and healthy. Symptoms included proptosis, periorbital and cheek swelling, loss of vision, diplopia, ecchymosis, and subconjunctival hemorrhage. The onset was commonly immediate. All patients had a CT scan (Figure 5), and two patients underwent emergent lateral canthotomy and inferior cantholysis [85,86], whereas one case opted for re-opening the incision for electrocautery of the bleeding vessel. All patients had an adequate resolution of symptoms. 

Although not recorded in any of the patients, the Valsalva maneuver during the procedure could have caused this ocular emergency by ultimately increasing jugular venous pressure to the orbit and subsequently rupturing periorbital-bridging vessels [84]. Additionally, bleeding from the posterior superior alveolar (PSA) artery or the pterygoid plexus to the inferior orbital fissure is a proposed route in atraumatic tooth extraction [84]. This theory is further supported by the fact that all three hematomas were localized to the extraconal space (Figure 5) or adjacent to the inferior orbital fissure [84,85,86]. 

Permanent vision damage was avoided with fast diagnosis and acknowledgement of its urgency. Due to its vision-threatening potential, retrobulbar hemorrhage should be kept in mind when performing dental extractions. In cases of small hematomas where vision is unaffected, conservative management with head elevation, steroids, and hyperosmolar agents can replace surgical decompression [84]. Finally, unintentional Valsalva maneuvers and uncontrolled high blood pressure should be avoided at the time of the procedure. This not only reduces the risks of retrobulbar hemorrhage but also of retinal hemorrhage. Indeed, a particular case of intraocular hemorrhage in a man with uncontrolled hypertension presented with the loss of central visual field in his right eye immediately after maxillary dental prosthetic implants. The patient reported intermittent breath-holding during the procedure, and an intentional Valsalva maneuver was performed post-procedure to exclude iatrogenic opening of the maxillary sinus. Valsalva, stress, the administration of epinephrine during the anesthesia process, and already high blood pressure resulted in a retinal hemorrhage, and the patient required pars plana vitrectomy [87].

### 3.6. Orbital Trauma and Injury

Finally, orbital trauma is a rare complication of dental procedures that can cause severe, permanent, and debilitating consequences. Four cases were identified: one case of shearing forces causing bone penetration [88] and three cases of unexpected anatomical variation [36,89,90]. The case of shearing forces involved the accidental penetration of the zygomatic bone during dental implantation, damaging the right lateral rectus. This patient ultimately remained with permanent esotropia despite interventions [88]. 

Having a solid understanding of the surrounding anatomy with detailed pre-surgical planning is a simple but indispensable way to avoid such complications. This is especially important in cases where anatomical variations are foreseeable. Cleft palate patients are particularly vulnerable to anatomical variations due to the multiple palate surgeries they undergo [90]. Cases of sphenoid fracture, hematoma, and ophthalmoplegia in cleft patients post-Le Fort I osteotomy have been documented [89,90]. However, unforeseeable anatomical variations can also cause important ocular complications. Babe et al. (2017) report a case of maxillary tuberosity fracture following a third molar removal, where the molar was abnormally ankylosed to the bone, causing excessive hemorrhage [36]. This case highlights how the fast reaction and stabilization of a patient reduces the chances of life-altering consequences. 

Through these unique cases of bony fractures leading to ophthalmic complications, the importance of pre-operative planning in all four cases and collaborative efforts between specialists remains key in avoiding potentially debilitating ophthalmic injuries. Three-dimensional quantitative analysis may also be a novel method used to identify abnormal anatomical variations, improving surgical planning [89].

## 4. Future Directions and Research Gaps

Dental procedures can cause ocular complications, notably neuro–ophthalmic, such as cranial nerve palsies and amaurosis. Although these complications are mainly benign and self-limiting, there are other conditions for which prompt recognition and diagnosis play a key role in disease progression. These include retrobulbar hemorrhages, orbital abscess, orbital emphysema, CRAO, and BRAO. If left untreated, hemorrhage, abscess, and emphysema can lead to orbital compartment syndrome, cavernous sinus thrombosis, and pneumomediastinum, respectively. The anesthetic injection process is the root cause of many complications. For instance, a case of facial nerve palsy was documented following an inferior alveolar block injected more posteriorly than is standard, causing direct trauma to the branches of the facial nerve [91]. Moreover, the possibilities of anatomical variations of arteries further increase the risk of unexpected intra-arterial injections. Indeed, the maxillary artery course and branching patterns are variable, serving as a possible unexpected target when performing a nerve block. Rare cases of the ophthalmic artery originating from the middle meningeal artery are documented in cadavers [41]. However, the process leading to ophthalmic complications may be multifactorial, and the influence of concomitant sinus or systemic infections, particularly in the context of orbital cellulitis, remains unclear. Perhaps active infections may warrant postponing dental procedures to reduce the risk of ophthalmological complications in the near future.

The prevention of these ocular complications requires a multidisciplinary effort. At the time of the procedure, correct visualization, stable placement of the anesthetic needle, two-plane aspiration followed by gentle injection, and awareness of nearby structures and common anatomical variations are essential to minimize the risk of damage to the neuro–vasculature and retrograde flow of an anesthetic agent to the orbit. If a similar clinical presentation occurs to a patient on more than one occasion, an anatomical variation is likely the cause and needs to be considered during future dental procedures [30]. This also applies to specific populations such as cleft patients, who are likely to have anatomical variations. Dental professionals should be aware of these ocular complications and their clinical presentation and recognize when timely referrals to an ophthalmologist or to the emergency department are necessary. This approach is vital because while most complications are self-limiting, some are irreversible and time-sensitive, especially in the context of ocular compartment syndrome. At this time, the patient should be assessed for emergency lateral canthotomy and cantholysis or decompression, respectively. Similarly, it is important for ophthalmologists and emergency physicians to recognize dental procedures as rare but possible causes of acute ocular pathologies. When a patient presents with ocular complaints following a dental procedure, it is crucial to obtain detailed information about the nature of the procedure. This includes identifying the type and technique of dental anesthesia used, determining the region of the oral cavity involved (e.g., maxillary or mandibular, as well as the specific tooth), and specifying the type of dental procedure performed. Such details are essential to better guide the differential diagnosis of ocular complaints and to establish a working diagnosis more efficiently. Despite the various strengths of this study, a few limitations should be noted. In fact, the low number of databases we used, that is, two, might not provide a complete overview of the ophthalmic complications that can arise from dental procedures. In addition, the broad scope of the topics limited our ability to explore each one in depth. Future studies focusing more precisely on individual topics are necessary to further examine them.

See Table 1 for a summary of each major ophthalmic complication of dental procedures, the anatomical explanation, common clinical features, and treatment. 

## Figures and Tables

**Figure 1 diseases-13-00244-f001:**
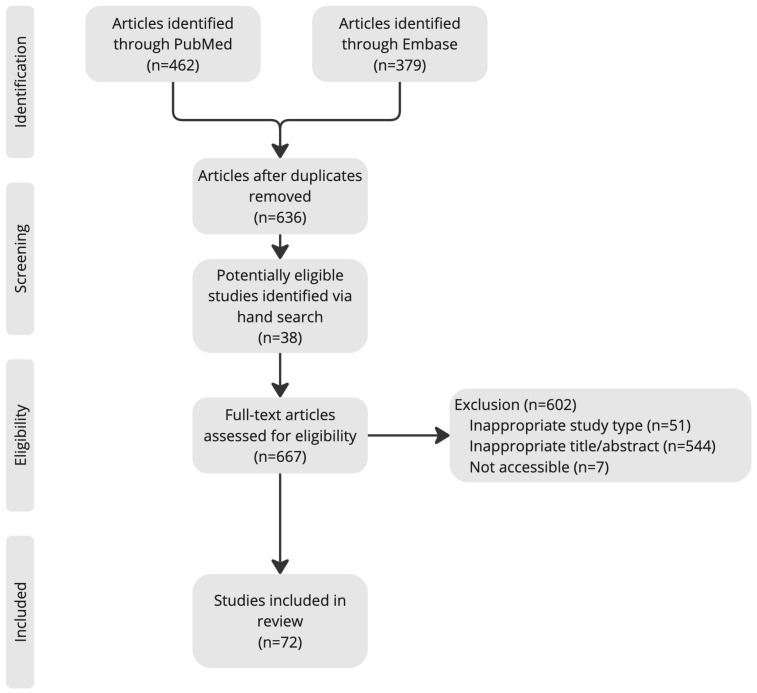
Article search and selection process.

**Figure 2 diseases-13-00244-f002:**
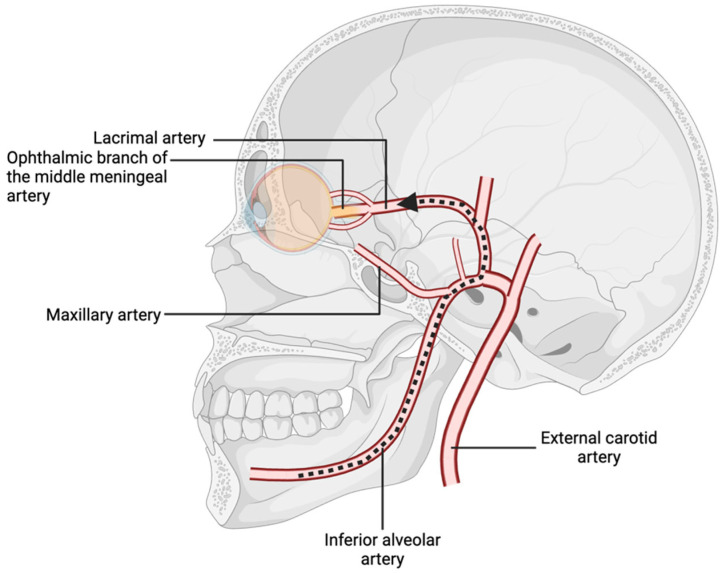
Branches of the internal maxillary artery, with pathway of retrograde flow from the inferior alveolar artery to the ophthalmic nerve.

**Figure 3 diseases-13-00244-f003:**
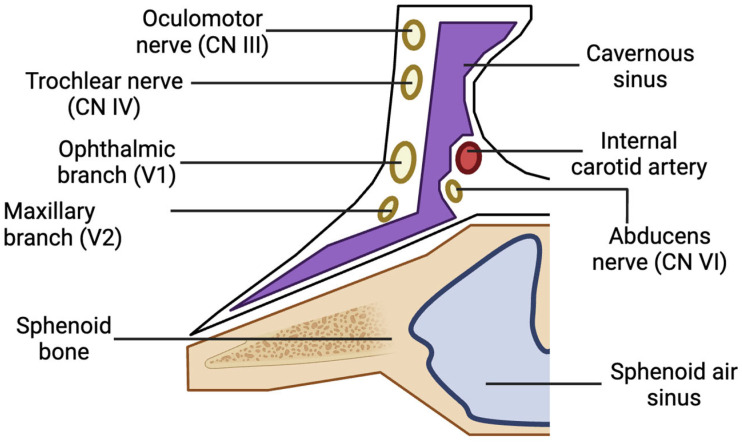
Anatomy of one side of the cavernous sinus and its contents.

**Figure 4 diseases-13-00244-f004:**
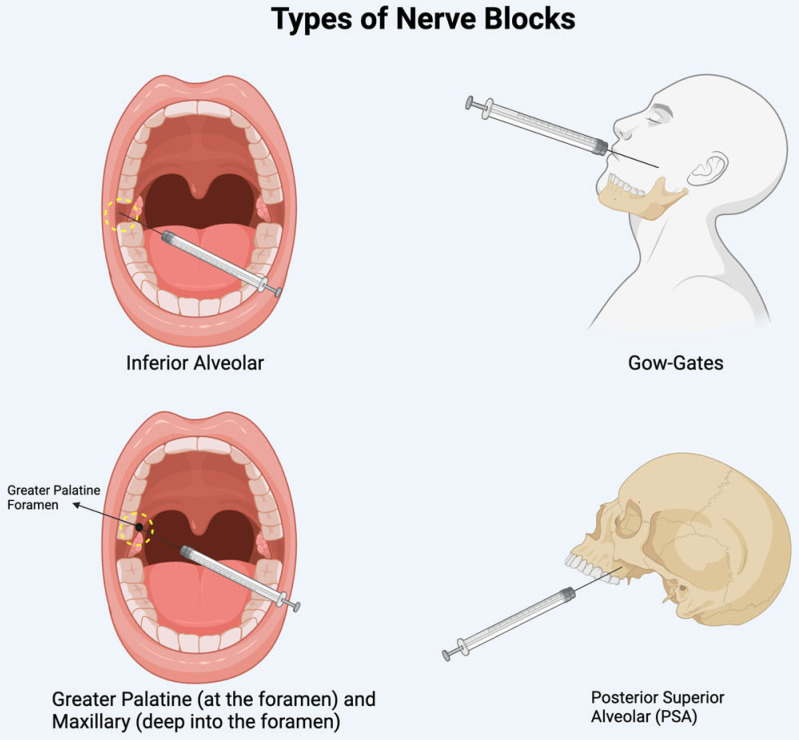
Illustration of the sites of nerve blocks discussed.

**Figure 5 diseases-13-00244-f005:**
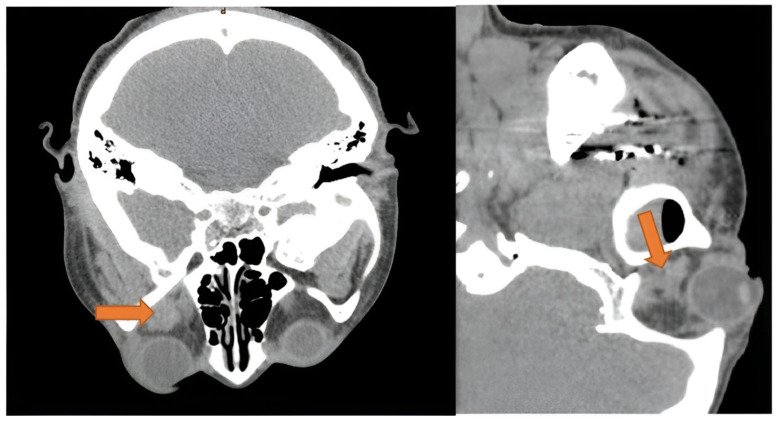
Axial and sagittal view of CT scan showing extraconal hematoma of the left orbit (orange arrow), with dimensions of 1.3 × 1.3 cm. Disclaimer: Reprinted with permission from ScienceDirect, copyright 2024, under a Creative Commons License, CC BY-NC-ND 4.0, https://creativecommons.org/licenses/by-nc-nd/4.0/ (accessed on 8 January 2025), Oral and Maxillofacial Surgery Cases, Vol 6, Suhaym et al. [85], “Retrobulbar Hemorrhage Following Tooth Extraction: Case Report & Anatomical Correlation”, https://www.sciencedirect.com/science/article/pii/S2214541920300031 (accessed on 8 January 2025); no changes made.

**Table 1 diseases-13-00244-t001:** Overview of ophthalmic complications of dental procedures.

Complication	Cases *	Anatomy	Features	Treatment
**Neuro–Ophthalmic**				
**Diplopia**	20	- Intra-arterial injection to the middle meningeal and ophthalmic arteries [11] - Direct diffusion to pterygomaxillary fossa and infraorbital fissure [22] - Infiltration of cavernous sinus [23]	- Double vision, predominantly due to abducens nerve palsy or lateral rectus muscle paresis	- Self -resolved - Cover affected eye
**Ptosis and loss of accommodation**	6	- Intra-arterial injection to the external carotid or posterior alveolar [37,41] - Direct diffusion to pterygomaxillary fossa [39] - Infiltration of cavernous sinus [38]	- Horner’s syndrome, ciliary palsy	- Self-resolved
**Vision loss**	19	- Intra-arterial injection to the middle meningeal artery, then to the ophthalmic artery [4] - Infiltration of cavernous sinus [47] - Retinal artery occlusion [8] - Ischemia of the optic nerve [49]	- Vision loss/low acuity - Flame-shaped hemorrhages, pale optic disk, cherry red spot	- Self-resolved or permanent vision loss
**Cervical Artery Dissection**	4	- Internal carotid artery - Oculomotor nerve	- Asymptomatic - Painful Horner’s syndrome - Orbital pain - Ipsilateral head or neck pain - Visual loss	- Strict bed rest with anticoagulation therapy
**Orbital and Periorbital Complications**				
**Orbital Cellulitis and Abscess**	11	- Connection between maxillary molar roots and maxillary sinus, then (1) through bony erosions [15,62] (2) directly to the inferior orbital fissure - Soft tissue spread (e.g., buccal cortical plate and periorbital tissues) [61] - Hematogenous spread by facial and ophthalmic veins [14,63]	- Proptosis, chemosis, limited EOM, visual acuity changes	- Antibiotics and surgical drainage - Emergency decompression if optic nerve affected - Left maxillary antrostomy, total ethmoidectomy [15]
**Subretinal** **Abscess**	1	- Hematogenous spread of pathogen from a primary extraocular source [66]	- Reduced visual acuity - Cells in anterior chamber and vitreous - Retinal abscess and white emboli branches	- Pars plana vitrectomy - Systemic and IV antibiotics
**Orbital Emphysema**	8	- Air from high-pressure tools or solutions travel through damaged mucosa of molars through fascial planes [70] or bony defects [72]	- Unilateral swelling and crepitus	- Self-resolved with time, ± antibiotics - Serial CTs and close follow-ups
**Periocular Necrotizing Fasciitis**	7	- Connection through deep fascial planes to the orbit and periocular structures [83]	- Pain - Swelling - Cellulitis progressing to necrosis	- Surgical debridement - Antibiotic administration - Post-operative surgery to correct complications or for aesthetic purposes
**Vascular Complications**				
**Retrobulbar Hemorrhage**	3	- Vasalva maneuver leading to rupture of ocular-bridging veins [84] - PSA a. or pterygoid plexus to the orbital fissure [84]	- Proptosis, swelling, - Loss of vision, diplopia, - Ecchymosis, subconjunctival hemorrhage	- Emergent lateral canthotomy and inferior cantholysis - Drain placement
**Subretinal Hemorrhage**	1	- Intentional and unintentional vasalva, epinephrine, uncontrolled HTN cause increased pressure of the retinal vessels [87]	- Reduced visual acuity, - Subhyaloidal and intraretinal hemorrhage, blood in vitreous	- Pars plana vitrectomy
**Others**				
**Ocular Trauma**	4	- Zygomatic bone is a component of orbital wall [89] - Maxillary tuberosity in close proximity to tributary vessels [36] - Proximity of sphenoid bone to cavernous sinus [89]	- Diplopia, changes in visual acuity, - Swelling and ecchymosis of affected area	- CT scan - Hemostatic agents for bleeding - Vacuum splint for bleeding [36] - ± steroids [36,89]
**Facial Nerve Palsy**	1	- Injection of anesthetic through the parotid gland to the peripheral branches of the facial nerve	- Unilateral facial weakness and inability to close ipsilateral eye	- Self-resolved

* Sorted by the main topic of each article.

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
