# Peer review of "Ophthalmic Complications After Dental Procedures: Scoping Review"

_diseases, 2025, doi:10.3390/diseases13080244_

Round 1
Reviewer 1 Report
Comments and Suggestions for Authors
Comments for the Authors
1) The current review article summarizes ocular complications associated with dental procedures. The review summarizes relevant anatomic connections between oral and ocular regions. The methodology includes published case reports and case series from 1950-2024. The authors have done an excellent job of summarizing these publications.
2) The methodology includes only published reports. Additional personal experience is not available.
3) The authors state that most of the ophthalmological complications following dental procedures are often reversible but some can be vision threatening or lead to permanent sequelae if not properly recognized and managed. These ocular complications following dental procedures are rare. The authors have provided excellent illustrations and tables to summarize these complications. In addition, they have included 86 references which provide an excellent background for this rare condition.
Author Response
Comment 1: The current review article summarizes ocular complications associated with dental procedures. The review summarizes relevant anatomic connections between oral and ocular regions. The methodology includes published case reports and case series from 1950-2024. The authors have done an excellent job of summarizing these publications.
Response 1: Thank you very much for taking the time to review our manuscript.
Comment 2: The methodology includes only published reports. Additional personal experience is not available.
Response 2: Thank you for your comment. To clarify, only case reports were included in this scoping review. There are very limited personal experience that the authors have encountered of cases of ophthalmologic complications after dental procedures, thus no personal experience was included. However, recommendations based in the case reports were provided in the "Future directions and research gaps" section.
Comment 3: The authors state that most of the ophthalmological complications following dental procedures are often reversible but some can be vision threatening or lead to permanent sequelae if not properly recognized and managed. These ocular complications following dental procedures are rare. The authors have provided excellent illustrations and tables to summarize these complications. In addition, they have included 86 references which provide an excellent background for this rare condition.
Response 3: Thank you again for taking the time to read our manuscript and provide us with your feedback and expertise.
Reviewer 2 Report
Comments and Suggestions for Authors Thank you for the thorough review. The authors address an important and underreported topic, and they provide valuable information about ophthalmic complications following dental procedures. I have some comments that may help improve the quality of the manuscript. 1. The manuscript is very lengthy, with some paragraphs exceeding 40 lines, which can make it hard for readers to follow the flow. I recommend the authors to consider summarizing some details and dividing long paragraphs into shorter subparagraphs. For example, in section 3.4.2 "Orbital and Subcutaneous Emphysema," it would be helpful to divide the content into separate paragraphs covering clinical manifestations and epidemiology, diagnostic methods and clinical course, and preventive strategies. 2. The authors tend to describe some case reports in excessive detail, such as the reports by Krauthammer (line 474) and Baba (line 482). The main goal of a review article is to integrate findings from various sources and highlight the most relevant information related to the topic, rather than providing in-depth descriptions of individual cases. Readers interested in more details can refer to the original sources. I recommend that the authors condense some of the case report descriptions to maintain focus and improve the flow of the manuscript. 3. The authors provide detailed information on pneumomediastinum and mediastinitis (lines 369–382). In my opinion, this content is not related to the main topic of ophthalmic complications and may distract the readers. I suggest either removing this section or summarizing it in 1-2 sentences. 4. Section 3.5.2 "Cervical Artery Dissection" is relevant to ophthalmic complications mainly through its association with Horner’s syndrome. I recommend mentioning cervical artery dissection as an etiology of Horner’s syndrome under the "Neuro-ophthalmic Complications" section and removing it as a standalone section. 5. I recommend the authors to include a separate section dedicated to preventive strategies discussing proactive measures to minimize the risk of ophthalmic complications following dental procedures. 6. I recommend the authors to include periocular necrotizing fasciitis as a serious potential complication following dental procedures, since several cases have been reported in the literature.Author Response
Comment 1: The manuscript is very lengthy, with some paragraphs exceeding 40 lines, which can make it hard for readers to follow the flow. I recommend the authors to consider summarizing some details and dividing long paragraphs into shorter subparagraphs. For example, in section 3.4.2 "Orbital and Subcutaneous Emphysema," it would be helpful to divide the content into separate paragraphs covering clinical manifestations and epidemiology, diagnostic methods and clinical course, and preventive strategies.
Response 1: Thank you for highlighting this. We have reviewed the section and have separated it into two shorter subparagraphs, while still keeping both topics in the same section. This is due to the discussion on the management and prevention strategies of orbital and subcutaneous emphysema being identical (lines 374-417). We have also gone through the rest of the manuscript and have shorted multiple paragraphs and created subparagraphs to improve overall flow. Changes made are highlighted in green and blue.
Comment 2: The authors tend to describe some case reports in excessive detail, such as the reports by Krauthammer (line 474) and Baba (line 482). The main goal of a review article is to integrate findings from various sources and highlight the most relevant information related to the topic, rather than providing in-depth descriptions of individual cases. Readers interested in more details can refer to the original sources. I recommend that the authors condense some of the case report descriptions to maintain focus and improve the flow of the manuscript.
Response 2: Thank you for your feedback and expertise. We have significantly shortened the discussions of the case reports and have focused on summarizing them without going into extensive details. We have significantly reduced the details for the Trauma section where Krauthammer and Baba were discussed (lines 595-617). We have also reworked the manuscript throughout to improve flow of the manuscript.
Comment 3:The authors provide detailed information on pneumomediastinum and mediastinitis (lines 369–382). In my opinion, this content is not related to the main topic of ophthalmic complications and may distract the readers. I suggest either removing this section or summarizing it in 1-2 sentences.
Response 3: Thank you for your feedback, we agree and have removed the paragraph and only mention them briefly in lines 405-406 to recognize them as potential complications.
Comment 4: Section 3.5.2 "Cervical Artery Dissection" is relevant to ophthalmic complications mainly through its association with Horner’s syndrome. I recommend mentioning cervical artery dissection as an etiology of Horner’s syndrome under the "Neuro-ophthalmic Complications" section and removing it as a standalone section.
Response 4: Thank you for your recommendation. We have reworked the paragraph to highlight its association with Horner’s syndrome and have put the “Cervical Artery Dissection” section in the neuro-ophthalmic complications section now (lines 312-340)
Comment 5: I recommend the authors to include a separate section dedicated to preventive strategies discussing proactive measures to minimize the risk of ophthalmic complications following dental procedures.
Response 5: Thank you for taking the time to review our manuscript and provide us with suggestions and your expertise. We agree with the comment and have separated the future directions section into two paragraphs. One summarizing the findings and areas of further research and one paragraph strictly discussing preventative strategies (lines 643-669). We have still kept them in the same bigger section as it is our concluding paragraph.
Comment 6: I recommend the authors to include periocular necrotizing fasciitis as a serious potential complication following dental procedures, since several cases have been reported in the literature.
Response 6: Thank you for your insightful recommendation. We find this topic very interesting and relevant. We have performed a search and review of the current literature. We have added our findings in a new section titled periocular necrotizing fasciitis (lines 419-536).
Round 2
Reviewer 2 Report
Comments and Suggestions for Authors
The authos revised the paper according to my comments.
Please correct some errors in spacing.
Author Response
Comment 1: Please correct some errors in spacing.
Response 1: Thank you for highlighting this. We had corrected errors in spacing within sentences, notably in lines 403,410,413.